# Systematic Analysis and Identification of Drought-Responsive Genes of the *CAMTA* Gene Family in Wheat (*Triticum aestivum* L.)

**DOI:** 10.3390/ijms23094542

**Published:** 2022-04-20

**Authors:** Dezhou Wang, Xian Wu, Shiqin Gao, Shengquan Zhang, Weiwei Wang, Zhaofeng Fang, Shan Liu, Xiaoyan Wang, Changping Zhao, Yimiao Tang

**Affiliations:** 1Institute of Hybrid Wheat, Beijing Academy of Agriculture and Forestry Sciences, Beijing 100097, China; wangdezhou84@126.com (D.W.); gshiq@126.com (S.G.); zsq8200@126.com (S.Z.); wangweiwei2398@126.com (W.W.); fangzhaofeng-hi@163.com (Z.F.); liushan_1202@sina.com (S.L.); 2The Municipal Key Laboratory of the Molecular Genetics of Hybrid Wheat, Beijing 100097, China; 3Hubei Collaborative Innovation Center for Grain Industry, Agriculture College, Yangtze University, Jingzhou 434023, China; 15821105046@126.com (X.W.); wamail_wang@163.com (X.W.)

**Keywords:** abiotic stress, drought, *CAMTA* gene family, wheat

## Abstract

The calmodulin-binding transcription activator (CAMTA) is a Ca^2+^/CaM-mediated transcription factor (TF) that modulates plant stress responses and development. Although the investigations of CAMTAs in various organisms revealed a broad range of functions from sensory mechanisms to physiological activities in crops, little is known about the CAMTA family in wheat (*Triticum aestivum* L.). Here, we systematically analyzed phylogeny, gene expansion, conserved motifs, gene structure, *cis*-elements, chromosomal localization, and expression patterns of *CAMTA* genes in wheat. We described and confirmed, via molecular evolution and functional verification analyses, two new members of the family, *TaCAMTA5-B.1* and *TaCAMTA5-B.2*. In addition, we determined that the expression of most *TaCAMTA* genes responded to several abiotic stresses (drought, salt, heat, and cold) and ABA during the seedling stage, but it was mainly induced by drought stress. Our study provides considerable information about the changes in gene expression in wheat under stress, notably that drought stress-related gene expression in *TaCAMTA1b-B.1* transgenic lines was significantly upregulated under drought stress. In addition to providing a comprehensive view of *CAMTA* genes in wheat, our results indicate that *TaCAMTA1b-B.1* has a potential role in the drought stress response induced by a water deficit at the seedling stage.

## 1. Introduction

Ca^2+^ regulates many aspects of an organism’s life cycle and functions as a signal messenger in response to environmental stimuli [1,2]. In plants, three major types of Ca^2+^-sensor proteins, including calmodulin (CaM)/CaM-like proteins, calcium-dependent protein kinases (CDPKs), and calcineurin B-like proteins (CBLs), are involved in the mechanism of Ca^2+^-dependent transcription regulation [3]. Additionally, Ca^2+^ can bind to and control certain classes of transcription factors (TFs) [4]. For instance, calmodulin-binding transcription activators (CAMTAs) are well-studied CaM-binding TFs first identified in tobacco and later determined to exist in all multicellular organisms [5,6]. CAMTAs regulate more than 1000 genes in plants and are linked to environmental cues, with differential responses to various signals and stresses [5,7], as assumed to occur in other unknown gene families that are not yet investigated in many varieties of wheat [8,9]. Notably, CAMTA proteins participate in the rapid response to stress by efficiently transducing calcium signals [10]. Genetics data show that plant CAMTAs function in plant resistance against abiotic stresses by regulating many downstream genes [11,12,13,14,15,16,17,18].

*CAMTA* gene families have been systemically characterized in several plant species [19,20,21]. For instance, *Arabidopsis thaliana* (L.) Heynh encodes six CAMTAs, which are all rapidly and differentially triggered by environmental cues such as heat, cold, salinity, drought, and hormones [7,14,22,23,24,25]. CAMTA1 improves drought tolerance in Arabidopsis and regulates stress-responsive genes enriched with CAMTA recognition *cis*-elements [24]. AtCAMTA3 functions in cold tolerance [14,23]. AtCAMTA2, in association with WRKY46, regulates the expression of *AtALMT1* and acts as a suppressor of salicylic acid (SA) biosynthesis–related gene transcripts [26,27]. AtCAMTA1 and AtCAMTA5 both contain the pollen-specific *cis*-element of AVP1 to enhance Arabidopsis *VPPase* (*AVP1*) gene expression and collaborate during pollen development [28]. AtCAMTA3 acts as a Ca^2+^-mediated transcription regulator in SA-mediated systemic immune response through the transcriptional expression of *NPR1* [29]. Galon reported that AtCAMTA4 responded to auxin and stresses [30]. AtCAMTA6 is crucial for Na^+^ homeostasis and salt stress tolerance [25]. Similarly, CAMTAs from several plant species have been implicated in abiotic stress responses with their expression patterns, including *Populus trichocarpa* Torr. & Gray ex Hook [31]; *Nicotiana tabacum* L. [32]; *Phaseolus vulgaris* L. [19,33]; *Zea mays* L. [34]; *Fragaria ananassa* Duchesne [35]; *Citrus sinensis* (L.) Osbeck [36]; and *Linum usitatissimum* L. [37]. However, transgenic plants of *Solanum lycopersicum* L. [38,39]; *Glycine max* (L.) Merr [40,41]; and *Brassica napus* L. [42] were found to be related to drought stress and hormonal response, but the genetic and environmental interactions are poorly understood. To obtain a deeper understanding, the mechanisms underlying CAMTA with gain-of-function and loss-of-function mutants are under investigation.

Wheat is an important cereal crop globally (https://www.fao.org/home/en (accessed on 8 May 2020)). Arable land is decreasing with increasing population and wheat demand, and the only way to solve the food shortage problem is to increase the yield of wheat per area of land through the advancement of science and technology [43]. Abiotic stresses influence the morphology and physiology of crops at vegetative and reproductive stages [44]. Approximately 50% of the total land used for wheat cultivation is affected by periodic drought [45]. Lesk reported that statistical models predicted a 10% reduction in cereal yield due to drought or extreme weather conditions [46]. To resolve the abiotic stress issue, the approach of breeding local wild genotypes should shift from classical breeding to more advanced molecular applications to successfully express desired genes under specific environmental conditions [47]. Plant breeders and researchers at CIMMYT (The International Maize and Wheat Improvement Center), IRRI (International Rice Research Institute), ICARDA (International Center for Agricultural Research in the Dry Areas), and ICRISAT (International Crops Research Institute for the Semi-Arid Tropics) are working to develop the drought tolerant cultivars of major cereals and leguminous crops [48,49]. Wheat has one of the largest crop plant genomes, which makes working with bread wheat challenging from a genetics, genomics, and breeding perspective [50,51]. The genome-wide identification of *CAMTA* genes has been performed in wheat [52]. However, a comprehensive atlas of wheat *CAMTA* genes expression in response to drought stress remains limited. In this study, we obtained 17 *TaCAMTAs* from wheat genomes and characterized the expression profiles of *TaCAMTAs* in response to abiotic stress. Compared to previous works that emphasized CAMTAs’ responses to stresses in other species, the results of this study provide potential candidates for crop abiotic stress tolerance improvement, especially against drought stress tolerance.

## 2. Results

### 2.1. Identification and Structural Analyses of TaCAMTA Genes in the Wheat Genome

We identified 17 *CAMTA* genes in wheat, including previously unknown sequences, by conducting Bio-Linux software analysis of wheat genomes using hidden Markov model (HMM) profiles. All 17 *TaCAMTA* genes were confirmed by searching for conserved domains via the NCBI-CDD database. The *TaCAMTA* genes were named according to their similarity to Arabidopsis *CAMTA* genes. The details of the *TaCAMTA* genes, including gene name and ID, ORF length, number of amino acids and introns, predicted molecular weight (MW), isoelectric point (pI), and grand average of hydropathicity (GRAVY), are presented in Appendix A. The length of predicted TaCAMTA proteins ranged from 138 (TaCAMTA5-B.2) to 1066 (TaCAMTA1b-A) amino acids, the molecular weight ranged from 16.21 (TaCAMTA5-B.2) to 119.32 (TaCAMTA1b-A.1) kDa, and pI ranged from 5.32 (TaCAMTA4a-D) to 9.57 (TaCAMTA5-B.2) (Appendix A).

The conserved domains of the CAMTAs, including a CG-1 DNA binding domain, a TIG domain, ankyrin repeats, IQ motifs, and CaMBD from the N terminus to the C terminus, were predicted in TaCAMTA proteins. TaCAMTA1b-A/B.1/D and TaCAMTA4a-A/B/D contained all of the conserved domains, whereas TaCAMTA1b-B.2 and TaCAMTA1a-A/B/D contained all the conserved domains except for the IQ domain; TaCAMTA4b-B/D and TaCAMTA5-A/B.1/D had no TIG domain (Figure 1B; Appendix A). All TaCAMTA proteins possessed at least four domains, except TaCAMTA5-B.2, which contained only a CG-1 DNA binding domain. TaCAMTAs contained a highly conserved domain structure and indicated that their function may be similar with multicellular organisms.

Similar tertiary structures can reflect similar biological functions. We, therefore, predicted the tertiary structures for members of the CAMTA protein family via the SWISS-MODEL and Phyre2 online servers. The results indicated that CAMTA proteins mainly comprise α-helixes and random coils (Figure 1C). Moreover, proteins in the same group were structurally similar, although some displayed β-sheets and certain variations in the random coils. Comparing the 3D structure and amino acid sequence of TuCAMTAs with TtCAMTAs, AetCAMTAs and TaCAMTAs showed that their core component appeared to be highly consistent, with all CAMTAs possessing at least four conserved domain structures, except for TaCAMTA5-B.2, which only had a CG-1 domain (Appendix A). The evolutionary relationships among *Triticum urartu* Thüm. ex Gandilyan, *Aegilops tauschii* Coss, *Triticum turgidum* L., and common wheat with CAMTAs by examining 3D structure analysis can help to explain the origin and evolution of gene families and how they possess biological significance during species evolution [9,53,54].

### 2.2. Chromosomal Location of TaCAMTAs

Seventeen wheat *CAMTA* genes were mapped on different wheat chromosomes using the available wheat genome annotation information. The chromosomal locations of these genes were determined using gene IDs from Ensembl Plants (Appendix A). There were 5, 7, and 5 *CAMTA* genes located on A, B, and D sub-genomes, respectively. Most genes were distributed in chromosomes 2A/B/D; the remaining chromosomes each contained one of the identified genes. By contrast, no *CAMTA* gene was found in chromosomes 1A/B/D, 5A/D, 6A/B/D, and 7A/B/D. Therefore, we conclude *CAMTA* genes were unevenly distributed on all wheat chromosomes, indicating that gene duplication events might have occurred in wheat chromosomes 2A/B/D during evolution.

### 2.3. Phylogenetic Analysis of TaCAMTAs

To further explore the phylogenetic relationships among various species, all CAMTAs were aligned using ClustalW, and the maximum likelihood (ML) method was used to construct a phylogenetic tree. All CAMTA proteins from *Arabidopsis thaliana*, *Oryza sativa*, *Zea mays*, *Glycine max*, and wheat were grouped into three subfamilies (sub.) based on their homology, except for *TaCAMTA5-B.2.* The classification resulted in 26 proteins in sub. I, 11 in sub. II, and 16 in sub. III (Figure 2). Sub. I was also the largest subfamily with CAMTA proteins from five different plant species. *TaCAMTA1a-A/B/D* were closely related to *OsCAMTA1a.1* and *OsCAMTA1a.2*, whereas *TaCAMTA1b-A*, *TaCAMTA1b-B.1*, *TaCAMTA1b-B.2*, and *TaCAMTA1b-D* were closely linked with *OsCAMTA1b.1* and *OsCAMTA1b.2*. Three 3:1 ortholog gene pairs with 100% bootstrap values were identified between wheat and rice: *TaCAMTA4a-A/B/D*, *TaCAMTA4b-A/B/D*, and *TaCAMTA5-A/B.1/D* were classified into *OsCAMTA4a*, *OsCAMTA4b*, and *OsCAMTA5*, respectively. Orthologous gene pairs of CAMTA were identified in monocots; no sister pairs of genes were identified in dicots. In summary, CAMTA genes from different species shared evolutionary divergence before the common ancestor of monocots and dicots.

### 2.4. Cis-Acting Regulatory Elements in the Promoter Regions of TaCAMTAs

DNA-binding transcription factors usually function by recognizing *cis*-elements in the promoter region of their target genes. *Cis*-elements related to stresses and hormones in plants were identified. Stress-related elements included MYB and MYC (both of them are drought-responsive elements), MBS (CAACTG, MYB binding site involved in drought inducibility), DRE/CRT (RCCGAC, dehydration- and cold-responsive element), and G-box (CACGTG, environmental signal-responsive element). Phytohormone-related elements included ABRE (YACGTGK, ABA-responsive element), AuxR (GGTCCAT, auxin-responsive elements), TCA element/SARE (TCACG, SA-responsive promoter element), and CGTCA and TGACG motifs (involved in MeJA responsiveness).

It has been reported that more stress-related *cis*-elements are located in the promoter regions of wheat *CAMTA* genes than other plant species [42,55]. Here, we screened for *cis*-elements in the 2.0 kb promoter regions of the regulatory sequences of wheat *CAMTA* genes. Most *cis*-elements shown in Figure 3, especially stress-related ones such as ABRE, MYB, MYC, G-box, and TGACG-motif, were enriched in the promoters of *TaCAMTAs*. In total, 38 ABRE, 24 MYB, 18 MYC, and 27 G-box *cis*-elements were predicted in *TaCAMTA* promoters, indicating that wheat *CAMTA* genes may be more involved in the plant’s response to drought stress.

### 2.5. Stage- and Tissue-Specific Expression Levels of TaCAMTA Genes

The determination of tissue-specific expression patterns of *CAMTA* genes could help us investigate their biological functions. We generated heatmaps of the expression levels of *CAMTA* genes in root, leaf, stem, and seed tissues using the WheatOmics platform, together with the iTAK and Expression Atlas databases. The results showed that the expression patterns varied between *CAMTA* genes. As shown in Appendix A, most orthologous genes in Arabidopsis, rice, and wheat exhibited similar expression patterns. The transcript levels of all the *CAMTA* genes were higher in the roots than in other organs, except for *TaCAMTA5-B.2*, *OsCAMTA1a.2*, and *OsCAMTA1b.1*. These results implicated that *CAMTA* family genes might have functional importance in root development.

We used qPCR to analyze the expression levels of *TaCAMTA* genes in the root and shoot of 2-week-old seedlings; in the root, stem, leaf, stamen, and spike at flowering during the reproductive stage; and in grains at 20 days after anthesis (during seed formation). Transcriptome profiling revealed diverse expression patterns, similarly to recent reports by Yang et al. [52]. Most *CAMTA* genes in wheat were predominantly expressed in shoots during the seedling stage, whereas higher expression levels were observed in leaves during the reproductive stage (Figure 4). The overall expression, measured as gene expression in all organs at different developmental stages, produced inconsistent results, suggesting that *TaCAMTA* genes may play distinct roles during different stages of plant growth and development.

### 2.6. Expression Profiles of Predicted CAMTA Sequences during Abiotic Stress

Extensive studies have shown that the expression of *CAMTA* genes responds to abiotic stress. To investigate the potential roles of *TaCAMTAs* in response to stresses and in the establishment of appropriate tolerance during the seedling stage, we analyzed the expression of 17 *TaCAMTA* genes in wheat seedlings subjected to salt, heat, drought, cold, and ABA stresses, as described in the Materials and Methods section. Our findings showed that all *TaCAMTA* genes responded to at least two stress-related treatments, with more than a two-fold change in expression. Under drought treatment, the expression of *TaCAMTA1a-A/D*, *TaCAMTA1b-A/D*, *TaCAMTA4a-A/B/D*, *TaCAMTA4b-B*, and *TaCAMTA5-A* first increased and then decreased with time. *TaCAMTA1b-B.1* reached a maximum expression of 100-fold to 200-fold, whereas *TaCAMTA1a-B*, *TaCAMTA1b-B.2*, *TaCAMTA4b-A/D*, *TaCAMTA5-B.2*, and *TaCAMTA5-D* were suppressed (Appendix A). Seedlings treated with NaCl showed no significant changes in the expression of most *TaCAMTA* genes, except for *TaCAMTA1b-B.1/B.2*, *TaCAMTA4b-D*, and *TaCAMTA5-A/B.1*, which increased to peak at 24 h post-treatment (Appendix A). Additionally, most *TaCAMTA* gene expression patterns exhibited significant differences when subjected to ABA, heat, and cold (4 °C) stresses (Appendix A). These results indicated that *TaCAMTAs* responded to stresses, especially drought (Figure 5). In seedlings, the rapid response of *CAMTA* genes to these external chemical and physical stimuli suggests that they participate in the cross-talk between multiple signal transduction pathways involved in stress tolerance. The relative expression levels in roots of two-week-old seedlings were normalized to the expression of the reference gene and set to 1 at 0 h.

### 2.7. TaCAMTA1b-B.1 Regulates Prominent Drought-Responsive Genes

A preliminary investigation was undertaken to investigate the functions of the *TaCAMTA* gene family in wheat that included the determination of expression patterns in seedlings subjected to stress conditions. Collectively, the data obtained suggested that *TaCAMTA* genes respond to various stresses, but especially to drought conditions. Transcripts of all *TaCAMTA* genes were modified under moderate dehydration during the seedling stage; the expression of *TaCAMTA1b-B.1* markedly increased following drought stress, emphasizing that *TaCAMTA1b-B.1* is a strong candidate to function in stress responses. We constructed overexpression and RNAi vectors and produced transgenic specimens in T2 generation to further understand the function of *TaCAMTA1b-B.1*. All transgenic lines were identified at the transcriptional level using qPCR (Figure 6B).

We examined the growth response of transgenic plants to drought stress. No significant difference in the phenotype was observed among the RNAi, overexpression, and wild-type (WT) plants before drought treatment. However, after 2 weeks of drought treatment, the overexpression plants had more green leaves than WT plants, whereas the RNAi plants did not grow as well (Figure 6A). The survival rate was higher in overexpression plants than in WT plants when exposed to drought conditions but lower in RNAi plants after re-watering (Figure 6C). These data demonstrate that the *TaCAMTA1b-B.1* gene positively regulates drought tolerance in transgenic wheat.

Plants with a high capacity for water retention can better survive drought or dehydration stress. The water loss from isolated leaves reflects, to a certain extent, the water status of the entire plant. During 0–24 h of a dehydration treatment, overexpression plants showed a low water loss rate (WLR), whereas RNAi plants had high WLR compared with WT leaves (Figure 6D). These results indicated that *TaCAMTA1b-B.1* promotes the plant’s ability to retain water under dehydration conditions. We also determined that the MDA contents were lower in overexpression plants but higher in RNAi plants than in WT plants, with or without exposure to drought (Figure 6E). These analyses confirmed that tolerance to dehydration increased in *TaCAMTA1b-B.1*-OE lines but decreased in *TaCAMTA1b-B.1*-RNAi lines.

The expression levels of drought stress-related genes, such as *MYB33*, *NAC69-1*, and *NAC69-3*, are used to assess the capacity of wheat plants to respond to drought conditions. Here, we measured the relative expression levels of these three genes in WT and transgenic plants grown under normal and drought conditions. The qPCR results showed that the expression levels of stress-related genes were upregulated in both WT and transgenic lines after drought treatment, but that they were significantly higher in overexpressed transgenic plants than in WT plants (Figure 6F). Therefore, we conclude that *TaCAMTA1b-B.1* may regulate stress-related genes to promote drought resistance in plants.

## 3. Discussion

### 3.1. Molecular Mechanisms of CAMTA Duplication

Gene duplication is one of the most important steps in the evolution of complex genes from simple ones [56]. A domain is a well-defined region within a protein that either performs a specific function within a protein [57] or induces modifications at the structural level via changes in nucleotide and amino acid sequence to influence the evolution of the protein [58]. Internal repeats often correspond to functional or structural domains within the proteins, and an exact correspondence exists between gene exons and the structural domains of the protein product [56]. The segments are structurally very complex (larger blocks composed of small units or modules of duplications), and the abundance of segmental duplications is one of the primary forces driving the evolution of genes with new functions. The association of these segments with chromosomal instability and the rapid evolution of new functions occur in tandem with duplication events and are documented as segmental duplications [59]. Bread wheat exhibits allopolyploidy (AABBDD) via hybridization followed by the merging of genomic content of diverging species, with gene expansion through polyploidization coming from whole-genome duplication and small-scale duplication [60]. Although the genome-wide identification of the *CAMTA* family has been completed in wheat [52], we identified 17 *CAMTA* genes in wheat and characterized two new members of the *TaCAMTA* family, *TaCAMTA5-B.1* and *TaCAMTA5-B.2*, located on chromosome 2B. All units of the exon-intron structure of *TaCAMTA5-B.1* and *TaCAMTA5-B.2* have merged their genomic content into *TaCAMTA5-A* or *TaCAMTA5-D*, and an analysis of genomic DNA sequences and amino acid sequences revealed that *TaCAMTA5-B.1* and *TaCAMTA5-B.2* may have divided from the previously existing gene *TaCAMTA5-B*, a *TaCAMTA5* homoeologous gene (Figure 1). Notably, it is clear that *TaCAMTA5* had orthologs in the A genome lineage (*Triticum urartu*), the D genome lineage (*Aegilops tauschii*), and tetraploid wheat (*Triticum turgidum*) (Appendix A). TaCAMTA5-B.2 only contains the CG-1 domain (for any protein to be characterized as CAMTA, the presence of the CG-1 domain is obligatory [7]); however, TaCAMTA5-B.1 has multiple characteristic domains except for the CG-1 domain (the CaMBD domain is implicated in the association of Ca^2+^-loaded CaM to CAMTAs [7]). A previous study has reported the evolution and diversity of the CAMTA family from six Chlorophyta genomes and 35 plant genomes and showed that not every gene contained both a CG-1 domain and a CaMBD domain [61,62]. This evidence indicated the probable function of CAMTA proteins TaCAMTA5-B.1 and TaCAMTA5-B.2 (Figure 1B).

We also explored whether *TaCAMTA5-B.1* and *TaCAMTA5-B.2* belong to the *CAMTA* gene family. We cloned the coding regions of the *TaCAMTA5* genes, fused them with the pGBKT7 and pGADT7 vectors separately, and transformed each of these constructs individually into the yeast strain Y2HGold. The transformants carrying each construct grew well on selective SD medium lacking tryptophan (SD/−Trp or SD/−Trp−Leu). Transformants carrying *TaCAMTA5-B.1* and *TaCAMTA5-B.2* did not grow on SD/−Trp−His medium, indicating that these genes had no transactivation activity (Figure 7A). In addition, TaCAMTA5-B.1 interacted with TaCaM, as confirmed by the yeast two-hybrid assay, whereas TaCAMTA5-B.2 interacted with neither TaCaM nor TaCAMTA5-B.1 (Figure 7B). These results suggest that TaCAMTA5-B.1 and TaCAMTA5-B.2 may be involved in the Ca^2+^-CaM-CAMTA binding model and provided valuable insight into CAMTA with respect to its molecular functions.

### 3.2. Basic Characteristics of CAMTAs in Wheat

Phylogenetic analysis provided insight into the evolutionary relationships among the CAMTA family members across species, as well as putative functional assignments. Most *CAMTA* genes possess a conserved structure and a similar function [52]. Based on our phylogenetic analyses (bootstrap value = 100%), six sister pairs of genes between wheat and rice were identified as ortholog genes, suggesting that the functions of these TaCAMTAs might be similar to those of CAMTAs in rice. No sister pair of genes was identified between monocots and dicots (Figure 2). A comparison of the *CAMTA* gene family among *Triticeae* species (e.g., *Aegilops tauschii*, *Triticum urartu*, *Triticum turgidum*, and hexaploid bread wheat) showed that *TaCAMTA1b* had no ortholog in the A genome lineage and that the 3D structure prediction, combined motif analysis, and sequence alignment exhibited similar patterns. Most CAMTA tertiary structures were homologous (especially in the A, B, and D sub-genomes) but structurally different in their progenitors (Appendix A). These results indicate that intraspecies originate from whole-genomic duplication events and chromosomal arrangements from multiple evolutionary lineages through the combination of differentiated genomes. The evolution of CAMTA family members diverges across plant species [40]. Recently, CAMTAs from plant species were reported to play essential roles in abiotic stress responses [21]. Gene expression profiling in silico and qPCR analyses showed significant species specificity, tissue specificity, and developmental stage specificity at the transcription level (Figure 4 and Appendix A). The 17 *TaCAMTAs* were expressed differentially in various tissues subjected to hormone treatment or abiotic stresses (Appendix A). Under drought stress conditions, *TaCAMTAs* were significantly expressed at all time points during the seedling stage. Moreover, several *cis*-acting elements known to be related to drought stress were discovered in the promoter region. These observations led us to hypothesize the involvement of *TaCAMTAs* in drought stress regulation.

### 3.3. TaCAMTA1b-B.1 Regulates the Expression of Stress-Associated Genes in Response to Drought

TFs regulate virtually every aspect of the plant’s response to different stresses. We previously demonstrated that *TaCAMTA1b-B.1* is orthologous to *AtCAMTA1*, which is involved in stress response in Arabidopsis. The *camta1* mutant showed drought sensitivity, and AtCAMTA1 regulates a broad spectrum of stress-inducible genes such as *RD26*, *ERD7*, *RAB18*, *LTP*s, *COR78*, *CBF1*, and *HSP*s [7,24]. In this study, we detected drought tolerance in *TaCAMTA1b-B.1* overexpression lines but drought sensitivity in *TaCAMTA1b-B.1* RNAi lines (Figure 6A). *TaMYB33*, *TaNAC69-1*, and *TaNAC69-3* overexpression in wheat positively regulated the drought stress response, and the water loss rate in transgenic plants was low compared with WT plants under drought treatment [63,64]. Similar results were obtained in our study; the expression levels of the drought-related genes *MYB33*, *NAC69-1*, and *NAC69-3* were compared between transgenic and WT wheat plants under drought or normal conditions. We speculated that *TaCAMTA1b-B.1* might improve the drought tolerance of wheat by regulating TFs related to drought stress through Ca^2+^-mediated signal transduction. Our results are consistent with those of previous studies, which reported that other *CAMTA* genes positively regulated drought stress tolerance [19,32,33,34,37,39,40,41,52]. There is much evidence to suggest that *TaCAMTA1b-B.1* may function to improve the plant’s ability to retain water under drought conditions. The results demonstrated that the *TaCAMTA1b-B.1* gene could help reveal the molecular mechanism by which wheat induces a response to drought stress at seedling stage. However, there is no evidence that suggests that *TaCAMTAs* are induced by the drought stress of adult plants. Our study has provided useful information and insights into the functional divergence of *CAMTA* genes, which is worthy of further investigation. *CAMTA* genes are an appropriate candidate to be exploited as a plant genetic engineering tool in future studies.

## 4. Materials and Methods

### 4.1. Identification of CAMTA Genes in Wheat

We identified all *CAMTA* genes in several complete genomes with HMMER 3.0 (hidden Markov model, HMM) online software, using the Hidden Markov Model (HMM) profiles of the *CAMTA* gene family (Pfam03859: CG-1; Pfam12796: ankyrin repeats; Pfam01833: TIG domain; and Pfam00612: IQ were obtained from Pfam (http://pfam.xfam.org/ (accessed on 10 May 2020)) as queries against whole-genome peptide sequences of *Arabidopsis thaliana* (At), *Oryza sativa* (Os), *Glycine max* (Gm), *Zea mays* (Zm), *T. aestivum* (IWGSC), *T. urartu* (ASM34745v1), and *A. tauschii* (Aet_v4.0) from phytozome database v12.1 (http://www.phytozome.net (accessed on 13 May 2020)) and Ensembl Plants (http://plants.ensembl.org/index.html (accessed on 14 May 2020)). All CAMTA proteins identified after this initial search were examined by manual curation for the functional protein domains using the Calmodulin Target database (http://calcium.uhnres.utoronto.ca/ctdb/ctdb/ (accessed on 16 May 2020)) [31] and NCBI Conserved Domain Search (https://www.ncbi.nlm.nih.gov/Structure/cdd/wrpsb.cgi (accessed on 20 May 2020)). The physicochemical properties of CAMTA proteins were calculated using the online ProtParam tool (https://web.expasy.org/protparam/ (accessed on 21 May 2020)).

### 4.2. Chromosomal Location and Sequence Analysis of CAMTA Genes

Each chromosomal position and gene length was extracted from the Ensembl wheat genome database (http://ensembl.gramene.org/Triticum_aestivum/Info/Index (accessed on 23 May 2020)) to confirm and indicate the locations of the *CAMTA* genes in the chromosome. Next, all data about the chromosomal position and length of *CAMTA* genes were uploaded to Map Gene2 Chromosome v2 (http://mg2c.iask.in/mg2c_v2.0/ (accessed on 24 May 2020)) to obtain specific chromosomal locations. The 2 kb upstream promoter sequences of all TaCAMTA transcripts were obtained from the Ensembl Plants database and submitted to PlantCARE (http://bioinformatics.psb.ugent.be/webtools/plantcare/html/ (accessed on 25 May 2020)) for sequence analysis and to predict and locate their *cis*-acting regulatory elements.

### 4.3. Multiple Sequence Alignment and Phylogenetic Analysis of CAMTA

The amino acid sequences from four plant species were aligned using ClustalW. The phylogenetic tree was constructed using the MEGAX software and maximum likelihood (ML) method with 1000 bootstrap replications. Furthermore, the phylogenetic tree was visually enhanced using EvolView (https://www.evolgenius.info//evolview/#login (accessed on 26 May 2020))

### 4.4. Gene Structure and Conserved Motif Analysis of Wheat CAMTAs

The CDS and DNA sequences of *TaCAMTAs* were obtained from Ensembl Plants, and the gene structures were analyzed online via the Gene Structure Display Server v2.0 (GSDS) (http://gsds.cbi.pku.edu.cn/ (accessed on 27 May 2020)). In addition, the protein domain structures were obtained using the Domain Illustrator software (http://dog.biocuckoo.org/ (accessed on 28 May 2020)).

### 4.5. Vector Construction and Plant Transformation

The coding sequence of *TaCAMTA1b-B.1* was amplified from leaf cDNA using the specific primers *TaCAMTA1b-B.1-F* and *TaCAMTA1b-B.1-R* to generate overexpression lines. Following sequence verification, the full-length CDS was introduced to the expression vector pUBI-GFP using an adaptor containing KpnI (NEB, Beijing, China) and BamHI (NEB, Beijing, China) digestion sites. RNAi lines were generated by cloning a 184 bp fragment amplified from *TaCAMTA1b-B.1* cDNA into the pCUB vector through BamHI restriction sites in the sense orientation and SacI (NEB, Beijing, China) restriction sites in the antisense orientation. The resulting plasmids were transformed into *Agrobacterium* strain EHA105 and subsequently introduced into the wheat cultivar Jingdong18 (J18). *Agrobacterium*-mediated transformation to obtain transgenic plants following the protocol described by Wang et al. [65,66]. The primers used for vector construction are listed in Appendix A.

### 4.6. Yeast Two-Hybrid Assay

Full-length CDSs of *TaCAMTA5-B.1*, *TaCAMTA5-B.2*, and *TaCaM* were amplified from the cDNA of J18 and inserted into either the pGBKT7 or pGADT7 vector, separately. TaCAMTA5-B.1, TaCAMTA5-B.2, and TaCaM were checked for self-activation activity. pGBKT7 and pGADT7 were used as negative controls. All plasmids were co-transformed into the yeast strain Y2Hgold (Takara, Beijing, China) and cultured on SD/−Trp/−Leu (Takara, Beijing, China) plates for 48 h. The transformants were transferred onto 5-bromo-4-chloro-3-indolyl-b-D-galactopyranoside (SD/−Trp/−Leu/−His/−Ade/X-gal) plates for blue color development. Primers used for vector construction are listed in Appendix A.

### 4.7. Plant Materials and Stress Treatments

Winter wheat cultivar J18 was obtained from Institute of Hybrid Wheat, Beijing Academy of Agriculture and Forestry Sciences (http://123.127.160.133/index (accessed on 25 April 2019)). Wheat plants were grown in hydroponic boxes containing sterile water in a suitable incubator (Panasonic, Ehime, Japan) for 2 weeks under the following conditions: 25 °C/12 h light and 15 °C/12 h darkness. The root and shoot of 2-week-old seedlings in the root, stem, leaf, stamen, and spike at flowering during the reproductive stage; and in grains at 20 days after anthesis (during seed formation), were used to analyze tissue specificity. Whole seedlings were subjected to different abiotic stresses and hormone treatments. Seedlings were subjected to salt (incubated in 200 mM NaCl), drought (incubated in 25% (*w*/*v*) PEG-6000), ABA (sprayed with 100 mM), and cold (grown at 4 °C) treatments for 0 h, 1 h, 2 h, 5 h, 10 h, and 24 h [67] and heat (grown at 40 °C) treatment for 0, 1/4, 1/2, 1, 2, and 4 h [68]. The shoot tissue was collected at each time point. No treated plants were used as controls.

Seeds were sown in pots containing soil for 2 weeks to study the response of wheat plants to drought stress (Approximately 12 seeds of each genotype were spread on six pots). The 2-week-old seedlings were subjected to drought conditions by withholding water for 14 days. Control plants were watered normally. After 14 days of drought stress, the plants were watered and allowed to recover for 10 days before observing their survival response. The water loss rate and MDA contents of the leaves were determined according to the method described by [69,70], and the experiment was repeated independently three times. All samples were immediately frozen in liquid nitrogen and stored at −80 °C for subsequent analysis.

### 4.8. RNA Isolation and Quantitative Real-Time RT-PCR

Total RNA was extracted using Trizol. First-strand cDNA was synthesized using the Takara RNA extraction kit (Cat. No. RR047A, Dalian, China) according to the manufacturer’s instructions. SYBR qPCR Master Mix (Code. Q311-02, Nanjing, China) was used to perform RT-qPCR in Multiplate™ 96-well PCR plates (Bio-Rad, California, USA). Reaction systems were prepared in 20-µL volumes as follows: 10 µL of 2× Taq Pro Universal SYBR qPCR Master Mix, 1 µL of forward primer, 1 µL of reverse primer, 2 µL of cDNA, and 6 µL of RNase-free water. Each sample was tested using three technical replicates to ensure the accuracy of results.

## 5. Conclusions

In conclusion, 17 *CAMTA* genes have been identified and characterized in wheat. Our current research has provided new information of *TaCAMTAs* and described two new members of the family (*TaCAMTA5-B.1* and *TaCAMTA5-B.2*). The responsiveness of *TaCAMTA* genes to various abiotic stresses and hormones during the seedling stage demonstrate that *TaCAMTAs* are mainly induced by the drought stress. In particular, *TaCAMTA1b-B.1* played an important role in the drought stress response induced by a water deficit at the seedling stage. Further studies are still needed to reveal the functional significance of *TaCAMTA* genes in wheat, an important global crop with large and complex genome.

## Figures and Tables

**Figure 1 ijms-23-04542-f001:**
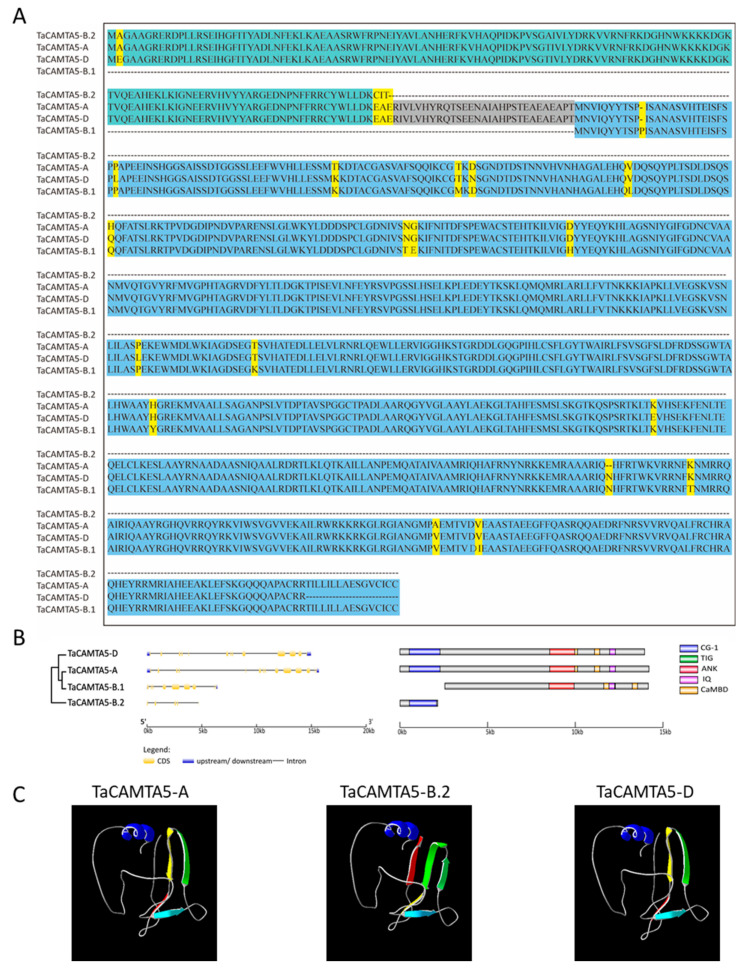
Sequence, domain and structure analysis of TaCAMTA5-B.1 and TaCAMTA5-B.2. (**A**) Alignment of TaCAMTA5-A, TaCAMTA5-B.1, TaCAMTA5-B.2, and TaCAMTA5-D sequences. (**B**) The structure and conserved domains of TaCAMTA5-A, TaCAMTA5-B.1, TaCAMTA5-B.2, and TaCAMTA5-D. (**C**) The predicted three-dimensional structures of TaCAMTA5-A, TaCAMTA5-B.2, and TaCAMTA5-D.

**Figure 2 ijms-23-04542-f002:**
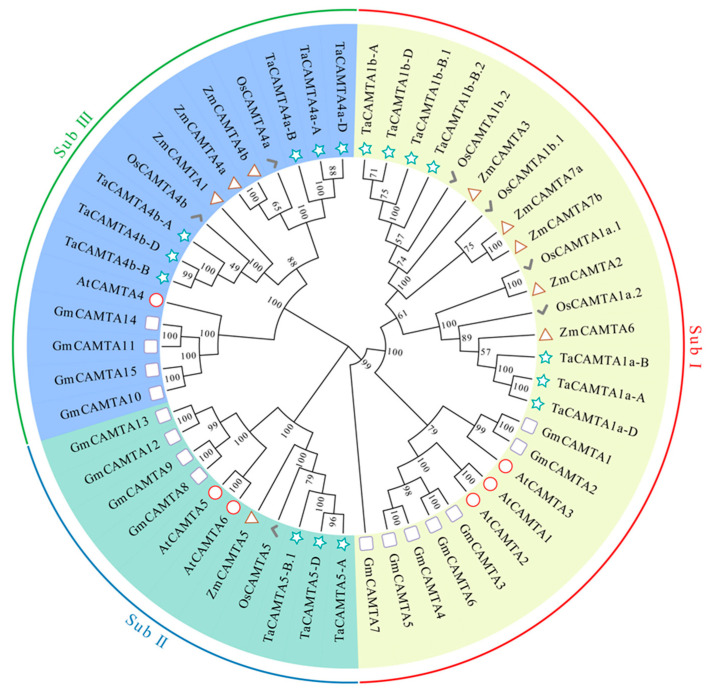
Phylogenetic relationships of CAMTAs in Arabidopsis and other crops. The maximum-likelihood (ML) phylogenetic tree was constructed based on the amino acid sequence alignments of *Arabidopsis thaliana* (6), *Oryza sativa* (7), *Triticum aestivum* (17), *Glycine max* (15), and *Zea mays* (9) using the MEGA X software with 1000 replicates. All CAMTAs are classified into three subfamilies, the names of which are shown in different colors outside of the circle.

**Figure 3 ijms-23-04542-f003:**
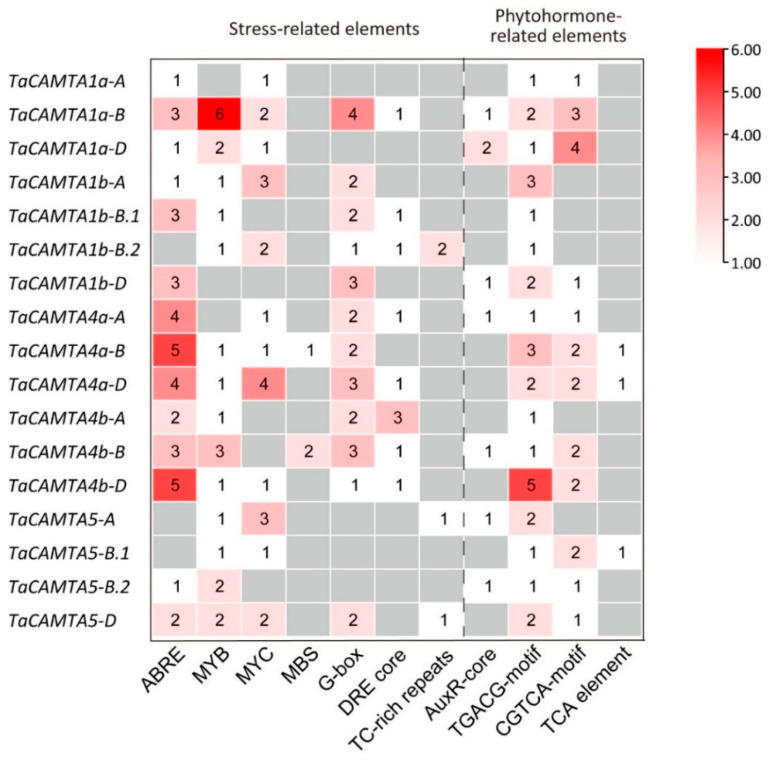
Abiotic stress and phytohormone response elements in the 2 kb upstream regions of *TaCAMTA* genes. The 2000 bp promoter regions of the corresponding *TaCAMTA* genes were used to analyze stress-related and phytohormone-related cis-elements; different cis-elements numbers are displayed with different colors. The gray color represents 0, with the white to red color gradually deepening to indicate an increment of 1 to 6.

**Figure 4 ijms-23-04542-f004:**
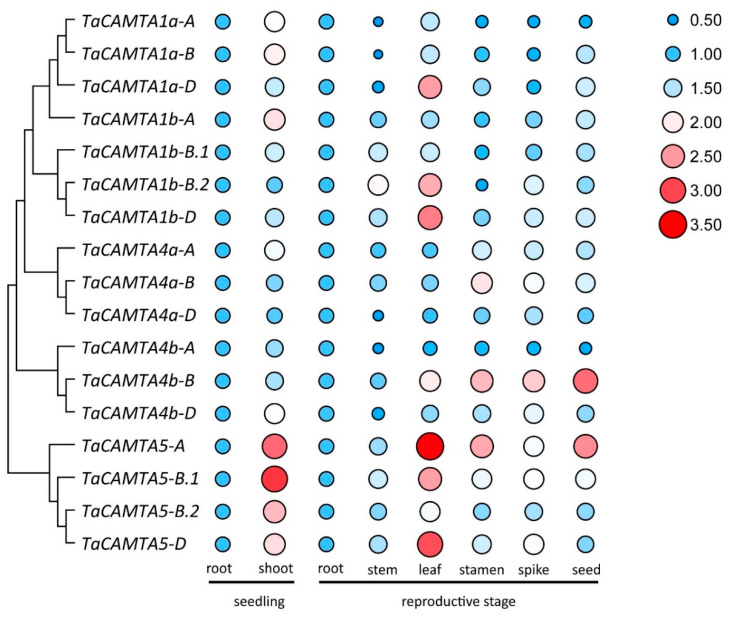
Quantitative real-time PCR analysis of *TaCAMTA* genes in different tissues during seedling and reproductive stages. The relative expression levels were normalized to 1 in roots. Data are presented as mean ± SD (*n* = 3).

**Figure 5 ijms-23-04542-f005:**
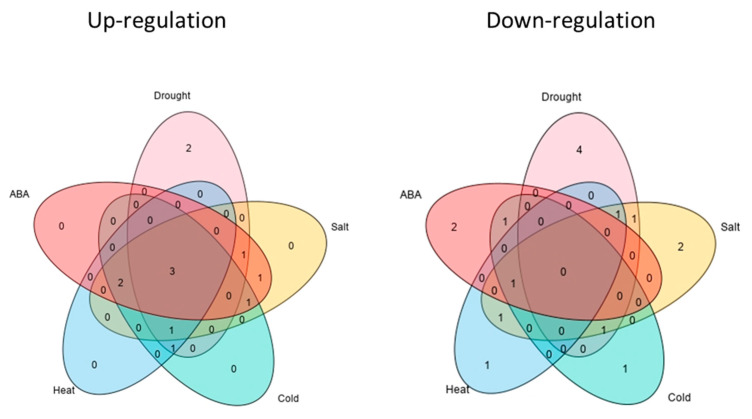
*TaCAMTA* transcript levels in response to drought, cold, NaCl, ABA, and heat treatments. Venn diagram for upregulated (**left**) and downregulated (**right**) *TaCAMTA* genes after exposure to different abiotic stresses or ABA treatment. Each section of the diagrams shows genes that are specific for a single stress (drought, salt, heat, cold, or ABA) or that are shared across multiple conditions.

**Figure 6 ijms-23-04542-f006:**
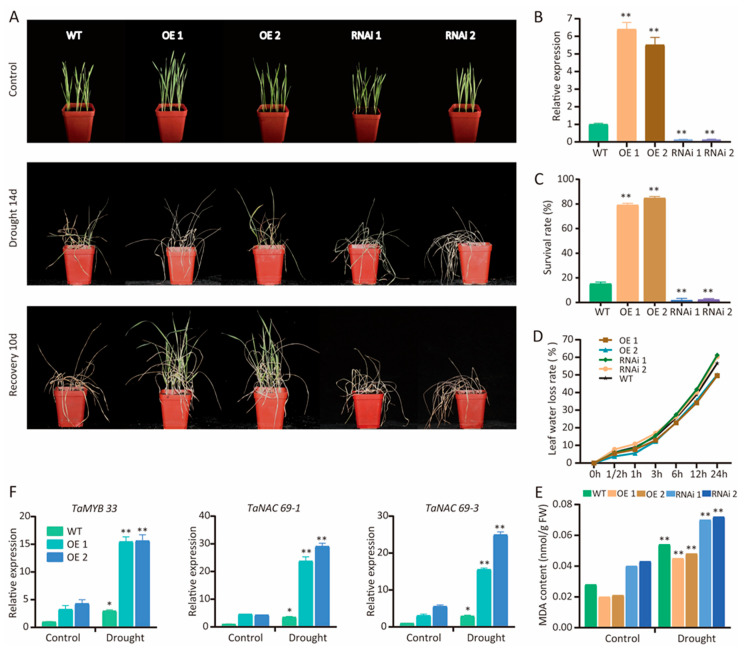
Performance of overexpression and RNAi in transgenic plants under drought stress in soil. (**A**) The transgenic plants were grown for 2 weeks under a regular water regime; water was then withheld for 14 days, after which plants were re-watered for 10 days. (**B**) Relative expression level of *TaCAMTA1b-B.1* in overexpression (OE) and RNAi cultivated specimens. (**C**) Survival rate of OE and RNAi plants. (**D**) Leaf water loss of OE and RNAi plants. (**E**) MDA content of OE and RNAi plants under the regular water regime and drought stress. (**F**) The expression level of known drought resistance genes in OE and RNAi-cultivated specimens. Data are presented as the mean ± SE of three different biological replicates and three technical replicates. The asterisks indicate a significant difference from control values at *p* < 0.05 (*) and *p* < 0.01(**), measured using the *t*-test.

**Figure 7 ijms-23-04542-f007:**
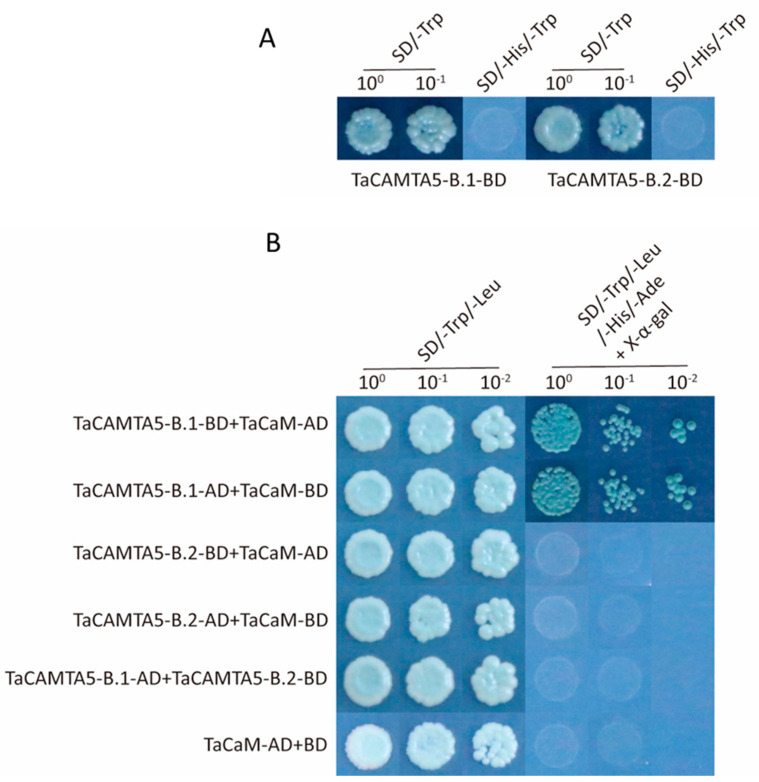
Y2H assay of TaCAMTA proteins. (**A**) Transcriptional activation identification of TaCAMTA5-B.1 and TaCAMTA5-B.2 proteins in yeast cells. (**B**) Y2H assay of interaction among TaCAMTA5-B.1, TaCAMTA5-B.2, and TaCaM in yeast cells. Full-length *TaCAMTA5-B.1*, *TaCAMTA5-B.2*, and *TaCaM* genes were fused with the pGBKT7 and pGADT7 vectors separately and subsequently expressed in the yeast strain Y2HGold. The transformed yeast cells were plated and grown on control (SD/−Trp, SD/−Trp−Leu) or selective (SD/−Trp−His and SD/−Trp−Leu−His−Ade + X-α-gal) plates. The plates were cultured at 30 °C, and photographs were captured 3–6 days after inoculation.

## Data Availability

Not applicable.

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
