# Peer review of "Systematic Analysis and Identification of Drought-Responsive Genes of the CAMTA Gene Family in Wheat (Triticum aestivum L.)"

_ijms, 2022, doi:10.3390/ijms23094542_

Round 1
Reviewer 1 Report
The authors propose a manuscript titled “Systematic Analysis and Identification of Drought-responsive Genes of the CAMTA Gene Family in Wheat (Triticum aestivum L.)”. The article give original information and is well written. The study were conducted on using the transcription activator named calmodulin (CAMTA). The study was conducted because there are few knoweldge about the effects and activities in physiological field of calmodulin in wheat (Triticum aestivum L.). The authors have systematically analyzed phylogeny, gene expansion, conserved motifs, gene structure, cis-elements, chromosomal localization, expression patterns of CAMTA genes in wheat and also have described two new members of the family (TaCAMTA5-B.1 and TaCAMTA5-B.2) Finally, have determined the expression of most TaCAMTA genes in response to several abiotic stresses (drought, salt, heat, and cold) and ABA during the seedling stage but was mainly induced by drought stress. The results indicate in particuylar that TaCAMTA1b-B.1 has a potential role in the droughtstress response induced by a water deficit at the seedling stage.
The manuscript deserve of few other crucial informations after which will be able published. There are no number lines in the draft of the manuscript? Why?
Introduction
Few remarks. Add some references and complete the crucial concept as follow suggested:
- The authors start the period with: “In plants, three major types of Ca2+- sensor proteins…”. Are you sure of the statement “in plants”? or in specific plant groups such as Poaceae?. Please clarify.
- This statement is correct but need a references. “CAMTAs regulate more than 1,000 genes in plants and are linked to environmental cues, with differential responses to various signals and stresses (Choose two references), as assumed occur in other unknown gene families not yet investigated in many varieties of Wheat (Perrino and Wagensommer 2022, Abenavoli et al. 2021)”;
- Please for botanical point of view is incorrect cited the scientific name without author name that discovered for the firt time the species. So, in this way and only for the first time in the manuscript please consider the scientific name in the complete form, as here suggested. Check whole documents. I suggested to use this website: http://ww2.bgbm.org/IOPI/gpc/query.asp
- Arabidopsis thaliana (L.) Heynh.
- Populus trichocarpa & Gray ex Hook.
- Nicotiana tabacum
- Phaseolus vulgaris…
- Zea mays…
- ….
- ….
Reference to be added:
- Abenavoli, L.; Milanovic, M.; Procopio, A.C.; Spampinato, G.; et al. Ancient wheats: beneficial effects on insulin resistance. Minerva Medica 2020 Doi: 10.23736/S0026-4806.20.06873-1
- Perrino, E.V.; Wagensommer, R.P. Crop Wild Relatives (CWRs) Threatened and Endemic to Italy: Urgent Actions for Protection and Use. Biology 2022, 11, 193. https://doi.org/10.3390/biology11020193
- Results
Well done, the figures and tables are clear. Few observations.
- Triticum urartu Thüm. ex Gandilyannot instead Triticum urartu、
- Aegilops tauschii instead Aegilops tauschii、
- Please complete this statement with 2/3 references “The evolutionary relationships among Triticum urartu Thüm. ex Gandilyannot, Aegilops tauschii and Triticum turgidumand bread wheat with CAMTAs throught 3D structure analysis can help to explain origin and evolution of gene families and possess biological significance during species evolution [choose 2/3 references. One of these: Perrino et al. 2014]”
Reference to be added:
- Perrino, E.V.; Wagensommer, R.P.; Medagli, P. The genus Aegilops (Poaceae) in Italy: taxonomy, geographical distribution, ecology, vulnerability and conservation. Systematics and Biodiversity 2014, 12, 331-349. Doi: 10.1080/14772000.2014.909543.
- Discussion
Well done, correctly referenced and written. The figure is clear. No observations.
- Materials and Methods
- The authors transparently declare the access date via web/internet, but no data was taken directly from the field? Is correct?. …Whole-genome peptide sequences were downloaded from the Phytozome database v12.1 (http://www.phytozome.net) and that the HMM profiles of the CAMTA domains (CG-1, ankyrin repeats, TIG domain, and IQ) were obtained from Pfam (http://pfam.xfam.org/), and CaMBD domain using the Calmodulin Target database (http://calcium.uhnres.utoronto.ca/ctdb/ctdb/). Finally, these domains were employed as a query to identify CAMTA gene family members using the hmmsearch program of HMMER. All candidate proteins were verified by submitting their sequences to the Pfam database and NCBI Conserved Domain Search (https://www.ncbi.nlm.nih.gov/Structure/cdd/wrpsb.cgi) platform to confirm the presence of the CAMTA domain.
- No field new data was considered in the study to to compare the data taken from the databases? Please clarify!
- Please give internet site or coordinates of the laboratory, because is too generic the name “The Municipal Key Laboratory of the Molecular Genetics of Hybrid Wheat, Beijing Academy of Agriculture and Forestry Sciences, Beijing, China”.
Conclusions
No conclusion? I suggest to writing two words on the aspect concerning the future researchs in this field
References
Please give doi number when available
Reviewer 2 Report
Manuscript Systematic Analysis and Identification of Drought-responsive Genes of the CAMTA Gene Family in Wheat (Triticum aestivum L.) by Dezhou Wang, Xian Wu, Shiqin Gao, Weiwei Wang, Zhaofeng Fang, Shan Liu, Changping Zhao, Xiaoyan Wang and Yimiao Tang contains some data on the two CAMTA Gene Family genes, a fairly comprehensive body of literature, and analyzes of the results. Unfortunately, it is not possible to understand how these genes are associated with resistance and in which cells and organs the expression also changes. With some degree of probability, it can be assumed that the expression changes and this suggests that these genes are associated with sensitivity. However, the situation is corrected by the relative knowledge of the homologues of these genes in other species. Perhaps this work may be useful in the future when studying the details of the study of tissue-specific expression of these genes. In its present form, this work is of a private nature but opens up some prospects for further development and can be published.
There are some comments on the design of the manuscript that require correction.
The last paragraph of the introduction should clearly articulate the goals and objectives of the work - this should be corrected.
Also, correction is required in the description of the methodology for setting up the experiment.
There is no information about repetitions, it seems that one pot was used and the experiment was done once - if this is not so, write about it!!!
The substrate, the volume of the pots, the age of the seedlings at the time of the start of the experiment, and how the treatments were done were not described. Why is it important? For example, if the cold was imitated while maintaining the lighting, the expression will be one, if without another. If you added 200mM NaCl, then how did you determine how much to add up to 80% moisture, up to 100% soil moisture capacity, poured from above, flooded (it is not clear), what is drought is not at all clear (it increased, moisture was maintained at a certain level (?) , ABA - too (watering, spraying, how the dose was determined per 1 plant per 1 m, for some weight) - one can only guess.
The very setting of the experiment on seedlings also raises questions. From the works of Rana Muntz of the last century, it is well known that the resistance and response of juvenile plants does not correlate with the response of adult plants. Meanwhile, drought and salinization often occur precisely at the stage of tillering and heading. In this case, in this small experiment, this is acceptable, but it must be taken into account in the conclusions that this is a special case, and not a characteristic of a general biological response, which should be reflected in the discussion.
Quite fantastic things are found in the description of the preparation of samples. What is a sample - it is not clear! They took leaves, took a growing point, took the main root, took adventitious roots? It is hard to believe that researchers of this level do not understand that there is a significant difference in expression not only at the cellular and tissue levels, but at least at the level of a plant organ. Correct this description as it was in reality.
Questions arise regarding transgenic wheat plants. Even now, the agrobacterial transformation of this culture remains an extremely difficult task. If you received transgenic plants yourself, you cannot be unaware of this. Requires confirmation of the absence of cantomination, confirmation of expression, transformation and regeneration protocols - none of this, if this work was done in another study or you received them - a link is required.
I think that without this information, consideration of the article is impossible. I hope the authors will correct the situation and supplement the materials with the missing data.
When discussing molecular aspects, plant physiology and cytology should not be ignored, as many works deal with stress damage to plants and a lack of understanding of the defense system at different levels makes such works less significant - expand the discussion of these aspects.
minor remark - there is no conclusion in the manuscript.
Round 2
Reviewer 1 Report
Dear authors,
I appreciate the work done following my suggestions.
This last version is able to be published.
Congratulation,
reviewer
Reviewer 2 Report
The manuscript "Systematic Analysis and Identification of Drought-responsive Genes of the CAMTA Gene Family in Wheat (Triticum aestivum L.)" has been significantly improved. I think that the work can be successfully published in its present form.